# The effect of mowing and mulching on snail communities: an experiment in wet meadows

Roland Farkas[1,2,3]*, Miklós Bán[3], Gergő Oláh[3], György Dudás[1], Zoltán Barta[3]

1 Bükk National Park Directorate, Eger, Hungary, 2 Juhász-Nagy Pál Doctoral School, University of Debrecen, Debrecen, Hungary, 3 HUN-REN-DE Behavioural Ecology Research Group, Department of Evolutionary Zoology and Human Biology, University of Debrecen, Debrecen, Hungary

* farkasro@yahoo.com

## Abstract

The condition of wet meadows nowadays depends mainly on human activities; the biodiversity and productivity of these habitats can only be maintained through appropriate management methods. Mowing and grazing are well-known traditional methods, but a new method, regular mulching – the shredding of plant material and its deposition in the area – is becoming more widely practised. To better understand the impact of these methods we directly compared their effects on snails, an invertebrate taxon common in these habitats. We experimentally manipulated two wet meadows in Northern Hungary, Europe, and surveyed their snail communities immediately before and fourteen months after treatment. Our results showed that mowing had a detectable negative effect on the snail communities, whilst mulching did not alter their characteristics. Therefore, mulching may be a promising candidate for conservation management in wet meadow habitats.

## Introduction

Grasslands, making up more than 40% of lands worldwide, are home to many specialist and endemic species [1]. Moreover, the development of human culture was also closely linked to grasslands. Initially it was used for nomadic animal husbandry and later for the rearing of livestock and growing forage [2]. Thus, in addition to their outstanding ecological value, grasslands are also highly important economically.

The extent and condition of grasslands have changed over time since the agricultural revolution, driven primarily by human activities [3]. While a few activities, such as deforestation, even create new grasslands, their extent has generally shrunk at an increasing rate with the intensification of agriculture and urbanisation. Furthermore, the emergence of new agricultural technologies has also greatly altered other characteristics of these habitats, such as landscape diversity and biodiversity [4]. As a result, the vast majority of historical grasslands have been transformed and cultivated by humans, mainly for rearing and grazing farm animals [5] and only small areas of pristine grassland remain. Many species historically endemic to grasslands

**Data availability statement:** All (gyepkezeles-21-22-final.qmd, gyepkezeles_kiserlet_csigaadatok_2022vel_kiegeszitve.tab, terepi_adatlapok_osszesitoje_kimutatashoz_3pont0.tab) files are available from the https://adattar.unideb.hu/ Dataverse data repository database (accession number(s) https://doi.org/10.48428/ADATTAR/ULAHSC).

**Funding:** This was prepared with the professional support of the Doctoral Student Scholarship Program of the Co-operative Doctoral Program of the Ministry of Innovation and Technology financed from the National Research, Development and Innovation Fund (Hungary). The study was supported by a National Research, Development and Innovation Office grant (NKFIH K138503, https://nkfih.gov.hu). Zoltán Barta was supported by National Research, Development and Innovation Fund grants (TKP2021-NKTA-32 and K138503, https://nkfih.gov.hu) (Hungary). Gergő Oláh was supported by the EKÖP-24-3-I-DE-131 University Research Scholaship Program of the Ministry for Culture and Innovation from the source of the National Research, Development and Innovation Fund (RH/1094-173/2024, https://kormany.hu/kulturalis-es-innovacios-miniszterium). This work benefited from services and resources provided by the EGI e-Infrastructure, with the dedicated support of ELKH-CLOUD. The funders had no role in study design, data collection and analysis, decision to publish, or preparation of the manuscript.

**Competing interests:** The authors have declared that no competing interests exit.

were, however, able to settle in and survive on cultivated grasslands. Consequently, in modern times the fate of grassland communities strongly depends on their proper management by humans, making this a critical conservation concern.

The existence of wet meadows requires their soil to be saturated with water for at least part of the growing season. This wetness creates conditions which significantly isolates the habitat from its surroundings, offering unique conditions for species with specific habitat requirements. Because of these specific hydrological conditions, wet meadows are highly sensitive to environmental changes and are negatively affected by both current climatic changes and farming [6]. Consequently, wet meadows are an especially vulnerable form of grasslands, which have suffered significant reduction in area across Europe over the past few centuries [7]. As a result, they are classified as threatened by the European Red List of Habitats [8].

In the absence of human intervention, natural succession processes usually take place on grasslands, resulting in spontaneous reforestation where the encroachment of shrubs and woody plants threatens the persistence of grassland communities and can also negatively impact farming activities [9]. A striking example of these changes can be seen in Central and Eastern Europe, where the political and economic changes of the late 1980's and early 1990's drastically altered the agricultural landscape. Large-scale changes in ownership and economic restructuring led to the regional abandonment of large grassland areas, many of which are still in this state today [10,11]. As grasslands harbour significant biodiversity and are important for farming [1], the generally-accepted and desirable management strategy is to maintain their current condition, rather than allowing them to return to an ancestral forested state.

Traditionally, the main methods of preventing spontaneous reforestation were grazing and mowing. More recently, the process of mechanised mulching has been developed to manage [12] and recultivate abandoned grassland habitats [13]. Unfortunately, mulching has several different meanings in English and the specific mulching method is often not clearly defined in scientific articles, e.g., [14,15]. This methodological ambiguity makes it difficult to understand the effects of mulching on biodiversity. For instance, one common definition of mulching is the process of spreading plant material harvested elsewhere, for example to protect economic crops (e.g., [16]). Another is when plant material cut (but not shredded) by a mower machine is left to decompose on the spot (e.g., [13,17]). Here, we use a third definition, whereby the standing vegetation on the meadow is shredded with a flail mulcher machine, with the resulting material spread evenly over the surface of the meadow (i.e., not removed). This definition is also used by many studies [9,18–23]. Regular mulching is commonly used for habitat restoration in nature conservation areas to remove shrubs and smaller trees from abandoned grasslands. One of the reasons for the spread of mulching is that it is more economical than traditional methods [13] as it requires only a one-time intervention, while cutting, drying, raking, baling and transporting the bales during traditional mowing require more investment.

Mowing and mulching can have various effects on the environment and its plant and animal communities. Direct effects from the moving parts of the machines cause

damage to both the flora and fauna [24,25]. Indirect effects arise from the associated changes to the physical and chemical properties of the habitat [26]. In this case, it is important whether the plant material is removed from the area or left to decompose in place. When vegetation is removed, direct irradiance can penetrate deeper, even to the surface of the soil. This can result in higher temperatures and hence stronger evaporation compared to similar habitats with closed vegetation [27,28]. On the other hand, covering the surface with cut plant material can result in lower temperatures [29] and higher soil humidity [30]. Mowing and mulching also changes the soil microbiome, influencing decomposition processes and the amount of available nutrients [20,31,32]. Biomass accumulated on the soil surface significantly increases the activity of decomposing organisms and enzymes in the soil [33].

The impact of management on grasslands depends greatly on the method used and the characteristics of the habitat concerned, and is an important research topic. Most work so far has looked at how management practices change plant communities, in particular their species composition, biomass or priority species groups. In dry, semi-arid, nutrient-poor meadows and mountain pastures, both mowing and mulching have generally been found to have a positive impact on maintaining species-rich vegetation [9,17,19,34,35] (. Limited evidence suggests that both mowing and mulching increase species richness in wet, alluvial habitats [13].

In contrast to our more detailed knowledge on the impact of mowing and mulching on flora, their effects on fauna, especially on invertebrates, has been less well studied. According to the sparse literature, mowing mostly has a negative impact on the abundance and diversity of some invertebrate groups (snails [36,37], less mobile spiders [38], dung beetles [39], orthopterans [40]. Mulching has also been shown to have a similar negative effect on several groups' abundance and diversity (butterflies [15], bees and wasps [22], insect larvae, flower visitors [23]). However, the impact of these management methods can be influenced by a number of factors, such as the timing and the frequency of treatments [23,26]. The literature suggests that, independently of habitat type, it is difficult to draw general conclusions about the effects of grassland management on animals because different taxa living in the same habitat have been observed to respond differently to the same treatment [14,41]. Consequently, we need more detailed studies targeted at specific groups to better understand the effects of grassland management practices.

Snails (*Gastropoda*) are common and widespread inhabitants of wet meadows [42]. They are significant actors in the decomposition processes [43]: they stimulate the activity of microbial detritivores by fragmenting organic materials, and their excretion of faeces and mucus production provides favourable conditions for microbial life [44]. Furthermore, the high water content of their bodies, their slow movement and limited dispersal ability make snail populations especially sensitive to environmental change. This makes them excellent subjects for studying the effects of environmental factors in general, and impacts of management practices in particular [36,45]. Nevertheless, the impact of mulching on snail communities is unexplored.

The aim of our study was to identify sustainable management methods for wet meadows by using snails as a target taxon group on wet meadows that had remained untouched for decades. We mowed and mulched portions of these areas using standard agricultural machinery, following a stratified experimental design. After fourteen months, we returned to collect data on the current state of the snail communities and investigated the effects of each treatment on their abundance and diversity.

Mowing removes biomass, while mulching leaves it in place, so we expected different effects from the two treatments. Both treatments are drastic interventions in habitats that have been abandoned for decades, so we also expected to find significant differences compared to the control areas. We chose the fourteen-months interval to investigate the effects of treatments because early recognition of adverse impacts is important to prevent valuable habitats from being treated with harmful methods in the longer term.

## Methods

### Study area

The experiment was carried out at two sites in Northern Hungary, Europe (Fig 1). The central WGS84 coordinates of the sites are the following: Szilvásvárad X = 48.1250, Y = 20.3920; Tarnalelesz X = 48.0908, Y = 20.1506. Both sites are located

in stream valleys of a hilly area, west of the Bükk Mountains. This area is characterised by high forest cover, with wet meadows occurring almost exclusively along the valley bottom watercourses. Annual rainfall in this area ranges between 600–700 mm and the average annual temperature is around 9°C [46]. In the valley bottoms meadow moulding soils have developed on clayey, loamy, sandstone bedrock [46]. In both sites, the habitats studied were non-tussock tall-sedge beds, with a dominance of *Carex acutiformis*. The characteristic associated species observed were *Lysimachia vulgaris*, *Lythrum salicaria*, *Cirsium oleraceum* and *Urtica dioica*. Both sites were flat and appeared to have a homogeneous vegetation structure. No bushes or trees were present in any of the sites. An important criterion for the selection of the sites was that they represent wet meadows that have not been farmed for the last 30 years. To verify this, a series of archive aerial photographs (www.fentrol.hu) were used.

### Experimental design

A sample area of 45 x 15 meters was marked out in each site. These were divided into 9 plots of 5 x 15 meters each. Both the edges of the sample area and the boundaries of the plots were marked with permanent stakes. Out of the 9 plots, 3 were mowed, 3 were mulched, and 3 remained untouched as controls. Treatments were assigned to the plots in a random order. Each plot was divided into 5 sections of 5x3 meters. In each of these, a sample was taken from a randomly selected point, resulting in 45 sampling points at each site (15 per treatment) (Fig 1). This gives n = 30 replicates per treatment for the whole study (2 study sites x 3 treatment plots per site x 5 samples per plot).

On June 28, 2021 (Szilvásvárad) and June 29, 2021 (Tarnalelesz), a malacological sampling took place at each sampling point before the start of management. Treatments were carried out a few days after this sampling, on July 1, 2021 at both sites. This is a typical time for farmers in the region to carry out farming activities on meadows. In the mowed plots, we cut the vegetation with a disc mower machine pulled by a tractor. The cut material was immediately removed from the surface using hand rakes. Mulching treatment was done by a flail mulcher machine pulled by a tractor to shred the

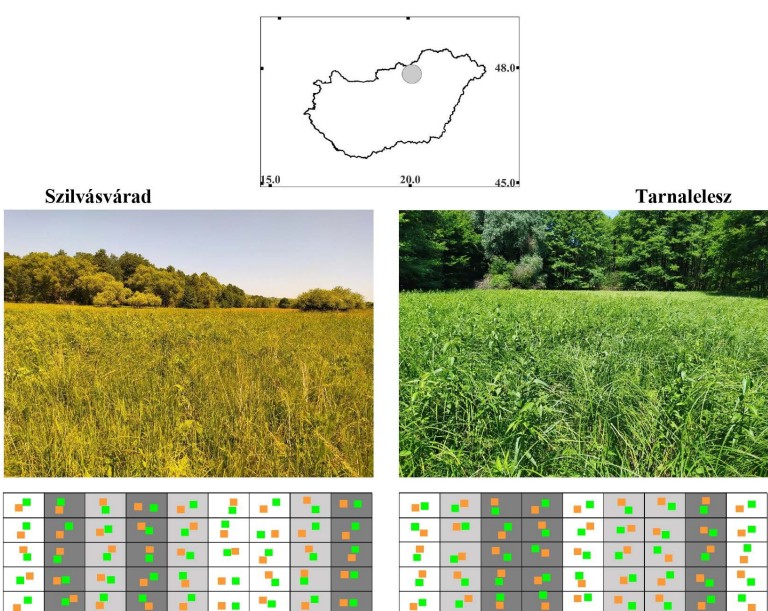

**Fig 1. The study sites.** Legend: Top centre: a map of Hungary with a grey circle indicating the location of the study areas. WGS 84 geographic coordinates are marked on the axes. Middle: photos of the habitats before the management. Bottom: detailed sampling design. Columns represent treatment plots, different colours indicate the type of treatment: white – control, light grey – mowed, dark grey – mulched. Randomly placed squares symbolize sampling points: orange – pretreatment, green – posttreatment. Photographs by Roland Farkas.

 

vegetation into pieces smaller than 10 cm. The resulting material was not removed and evenly covered the surface. Both sites were then left undisturbed. Subsequently, on August 15, 2022 (Szilvásvárad) and August 14, 2022 (Tarnalelesz), we went back to the sites and again collected a malacological sample from each sampling point. Malacological sampling was carried out in the morning hours each time. Sampling dates were selected to ensure that the weather conditions were similar and rain-free during the two-day periods.

One malacological sample was taken per sampling point on each occasion. Sampling was performed by collecting the leaf litter and the top 1 cm thick layer of the soil from a 25 x 25 cm quadrat. The collected samples were stored in airtight plastic bags in a cool, dark place. On the second day after collection, samples were floated: they were poured individually into buckets containing a volume of water significantly greater than the sample volume and left for half an hour. The floating material was then washed thoroughly in the buckets to remove any trapped snail shells and discarded. The material remaining in the bucket, including the snails, was washed through a sieve with a mesh size of 0.75 mm. The hole size was chosen so as not to allow the passage of the smallest adult snail specimens. The material remaining on the sieve contained the washed snail shells and other small organic and inorganic debris. This was immediately placed in 70% ethanol and stored. Later, the snail shells were removed from these samples using a 3.5X magnifying glass. The sorted snail shells were identified to the species level using a stereo microscope (Motic SMZ-168 series, 10X magnification). All determinations were carried out by RF, and they were based on the works of [42,47,48]. Only fully developed and intact (not broken or incomplete) shells of live adults were counted in the study. Live specimens were considered to be those in which the body of the snail could be observed inside the translucent shell or by looking into the aperture of the specimens. Slugs were not included in the study because this sampling method is not suitable for their quantitative analysis [49].

## Statistical analysis

All statistical analyses were performed in the R statistical environment [50]. First, we took a community based approach by calculating the Bray-Curtis distance matrix between the sampling points by the vegdist function of the vegan package (ver. 2.6–8, [51]) and then partitioned the variance of the distance matrix by performing a PERMANOVA [52] with sampling time (pre- and posttreatment), treatment (control, mowing and mulching) and their interaction as explanatory terms (adonis2 function of vegan). If the different treatments differently affect the changes in snail communities over time one would expect a significant interaction between the variables of treatment and sampling time. PERMANOVA performs random permutations of the data to derive the distributions of estimated parameters. In our model, spatial non-independence (i.e., data come from two separate areas) was controlled for by restricting permutations to within areas only [53].

Snail communities were further characterised by three community characteristics: (i) the total number of specimens, (ii) the number of species and (iii) Pielou's species evenness, $H' = \Sigma p_i \ln p_i / \ln S$, where $p_i$ is the relative abundance of species $i$ in the sample and $S$ is the number of species in the sample [54]. We fitted three generalised linear mixed effects models (GLMMs) to our data, one for each community characteristic. We entered treatment, sampling time and their interaction as fixed effects. We also took into account the statistical non-independence caused by spatial proximity by entering random effects into the models. Random effects were entered as sampling point ID nested within plot ID nested within site ID. Note, that we had to remove the random effect terms from the model for evenness, because this model did not converge as the variance explained by the random effect terms was essentially zero. We used a negative binomial error distribution for the number of specimens, and normal error distributions for the number of species and evenness. Inspecting the model fits with the DHARMa R package (v0.4.6, [55]) did not indicate any deviations from model assumptions.

We also analysed how treatments affected the number of individuals for the three most common species in the samples (*Carychium minimum*, *Vertigo angustior* and *Vallonia pulchella*). We fitted separate GLMM models for each species data with the same fixed and random effects as specified above. We used a negative binomial error distribution for all three species. Model assumptions were also checked by the DHARMa package.

All GLMM models were fitted with the 'glmmTMB' function of the glmmTMB package (v1.1.7, [56]). If a significant interaction was detected (by log ratio tests) we tested posthoc which treatment groups were significantly different from each other by the emmeans function of the emmeans package (v1.8.4-1, [57]).

## Results

During the study, 5233 living individuals of 20 species were found in the two sample sites combined (Fig 2). Total number of individuals in different study sites, treatment plots and sampling times is shown in S1 Table.

PERMANOVA analysis indicated a significant interaction between sampling time and treatment ($F_{2,169}=2.209$, p=0.003, $R^2=0.023$) suggesting that communities changed differently with different treatments. Investigating the fitted parameters of the PERMANOVA model revealed (i) no initial differences between the communities of different treatments ($F_{2,169}=1.033$, p=0.251, $R^2=0.010$), (ii) significant changes with sampling time ($F_{1,169}=17.619$, p=0.001, $R^2=0.091$) and that (iii) only the mowed community changed differently from the control ($F_{1,169}=2.921$, p=0.006, $R^2=0.015$).

Analysing community characteristics (abundance, species richness and evenness) with GLMMs, there were no significant differences between the mowed, mulched and control plots before treatment in any characteristic (S2 Table). However, we found a significant interaction between treatment and sampling time for all community characteristics: number of living individuals ($\chi_2^2=6.99$, p=0.03), species number ($\chi_2^2=11.32$, p=0.01) and evenness ($\chi_2^2=6.26$, p=0.04). This indicates that different treatments affected the snail communities differently.

Next, by comparing treatment groups, we examined how snail populations changed due to the treatments (Fig 3, S3 Table). The estimated number of living individuals decreased in all three management types, but this decrease was only significant in the mowed areas, indicating that the mowing itself likely caused a decrease. In terms of species numbers, again only mowing caused a significant decrease. For evenness, we also found a significant difference only in the mowed areas, but in this case the evenness increased, meaning that the abundance of each species in the community became more similar.

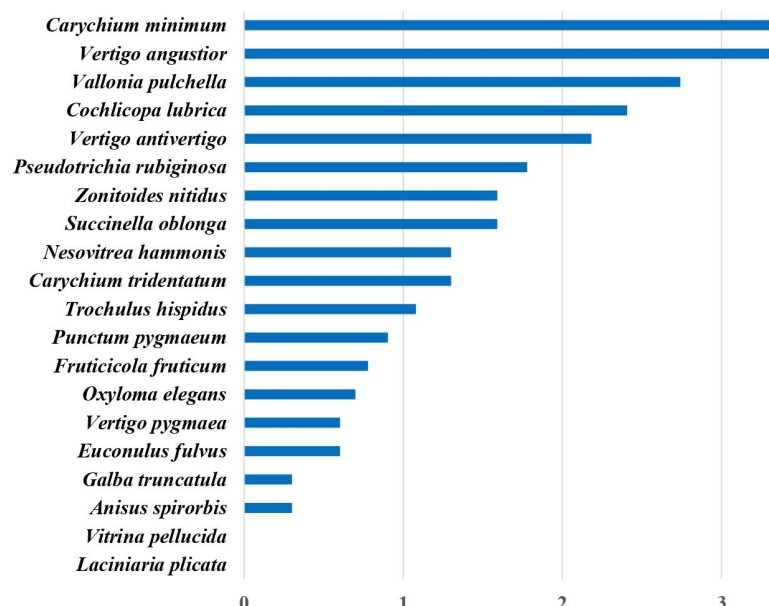

**Fig 2. Total number of living individuals in the samples.** Numbers of individuals are plotted on a logarithmic scale. Species are shown with decreasing numbers of individuals from top to bottom.

**Fig 3. The impact of treatment on three community characteristics.** The impact of treatment a) on the number of living individuals, b) on the number of species and c) on evenness. Vertical error bars indicate standard errors of the estimates derived by the emmeans function from the fitted GLMM models (***: $p < 0.001$, *: $p < 0.05$, ns: $p > 0.05$).

Finally, in addition to examining entire communities, we investigated whether treatments affected the abundance of the three most common species (*Carychium minimum*, *Vallonia pulchella*, *Vertigo angustior*). In *C. minimum* the abundance decreased significantly with sampling time ($\chi^2_1 = 97.16$, $p < 0.001$) independently of treatment (effect of treatment: $\chi^2_2 = 0.72$, $p = 0.696$; interaction: $\chi^2_2 = 2.85$, $p = 0.241$). In the case of *V. pulchella* neither treatments ($\chi^2_2 = 3.55$, $p = 0.169$), sampling times ($\chi^2_1 = 2.62$, $p = 0.105$), nor their interaction ($\chi^2_2 = 3.98$, $p = 0.137$) significantly affected abundance. Only for *Ve. angustior* did treatments have significantly different effects on abundance (interaction: $\chi^2_2 = 7.90$, $p = 0.019$); mulching significantly increased the number of individuals (posthoc test: $z = -3.25$, $p = 0.001$) while the other two treatments did not have such effect (control, posthoc test: $z = -1.53$, $p = 0.127$; mowing, posthoc test: $z = -0.81$, $p = 0.417$; Fig 4).

## Discussion

We found that snail communities declined significantly in mowed, but not in mulched areas, indicating that mowing can have a detectable negative impact on snail communities. This seemingly contradicts our finding that Pielou's evenness increased on mowed areas. However, the results are not necessarily mutually exclusive, since under certain conditions, diversity indices can increase despite a decrease in number of individuals and species. For instance, if the decrease is larger in more abundant species the distribution of individuals among species becomes more uniform and evenness increases. In cases like this, we must define our optimal outcome, and it seems sensible to choose a smaller decrease in abundance as more optimal than an increase in evenness. So in this situation, our finding remains, that mowing has negative effects on snail communities whereas mulching does not.

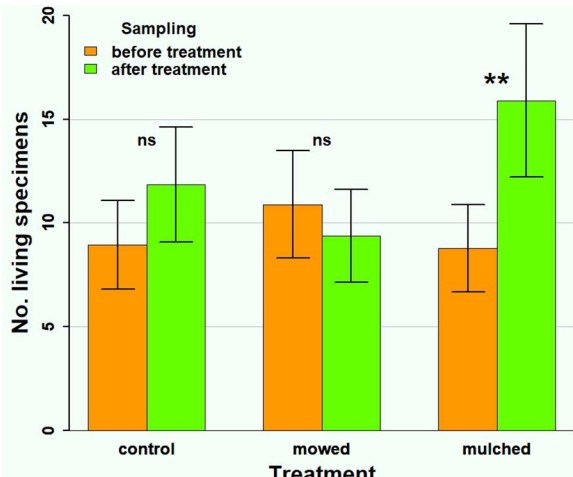

**Fig 4. The impact of treatment on the number of living *Vertigo angustior* individuals.** Vertical error bars indicate standard errors of the estimates (**: $p < 0.01$, *: $p < 0.05$, ns: $p > 0.05$).

There has been relatively little research on the effects of mowing on snails, and, as far as we know, there is no previous work regarding the effects of mulching. Previous research has mainly focused on longer-term effects over several years. For instance, Pech and colleagues ( [36] working in oligotrophic grasslands), and Farkas and colleagues ( [37] on valley-bottom wet meadows) found long-term reductions in abundance and species numbers due to mowing, but Wehner and colleagues ( [45] over a wide range of meadows) found an increase in snail community diversity in mowed areas, while no difference in abundance was found. All three studies are similar in that they were conducted on areas that were previously mowed (albeit for varying durations), while our current study was conducted in wet meadows that were previously untreated for decades. In addition, the above studies investigated the effects of mowing not by comparing samples taken before and after treatment on the same site, but by analyzing data from mowed and adjacent unmowed areas. Despite these methodological differences, our result on the detrimental effects of mowing is similar to two of these previous studies.

Our findings pertain to snail communities, yet wet meadows harbor numerous other abundant invertebrate taxa, which may respond differently to different management practices. Mowing had negative effects on abundance of less mobile spiders [38], species richness [41] and diversity [40] of orthopterans, and biomass of dung beetles [39]. Other studies have found that mulching decreased abundance of herbivorous insect larvae, flower visitors [23] and herbivore-hunting wasp [22] communities. Pižl and Starý [14] conclude that abundance of earthworms is lower on mowed than mulched sites, Schmitt [18] found that the abundance of butterflies is higher in mowed than mulched plots. However, the number of studied invertebrate groups is limited. This makes it difficult to draw general conclusions about the effects of treatments on entire invertebrate communities in a particular habitat. Moreover, the range of habitats studied is broad, with varying initial conditions: some studies monitor the effects of different ongoing treatments [18,39,40], while others established new experimental setups on habitats that had earlier undergone some form of regular treatment [22,23]. However, there are no previous studies where the initial state of the habitat, like ours, was untreated for decades.

Our results indicate that members of snail communities did not respond uniformly to the management. Of the three most common species we examined, both *C. minimum* and *Ve. angustior* are limited to wetland habitats, but only the latter was affected by the management. *V. pulchella* has a wider range of habitat preferences, and was also unaffected by the treatment. Similar results were observed in management studies with other taxa. Cizek and colleagues [41] found that the abundance of different ground beetle species changed differently with the same treatment. Pižl and Starý [14] showed

differential changes in abundance for different species of oribatids living above the soil surface in response to management. Both sources agree that generalist species tolerate changes better than specialists. *Vertigo angustior* is still considered a common species in Hungary, but its distribution in Western Europe has drastically declined, which can be attributed to changing and increasingly intensive agriculture [58]. Therefore, the appropriate choice of management practices is of paramount importance for its preservation.

The observed consistent declines in abundance of snail communities at both sites for all treatments were likely due to historic drought and heatwaves occurring in Hungary in 2022 [59]. In addition to these changes, management methods can have both direct (e.g., caused by the machines themselves) and indirect effects (e.g., caused by habitat transformation) on invertebrate communities. In our case, the changes observed in the mowed snail community are most probably due to indirect effects. The majority of the snail species studied live in the litter layer of the habitat, and the rotating parts of the equipment do not affect this layer. Being driven over by tractor wheels also has a direct negative effect on species living on the ground [24] but in our study the mowed and mulched plots were driven over to the same extent. Instead, the significantly higher decrease in abundance from mowing might have been mainly due to effects such as the sudden increase in irradiance and temperature, and a decrease in humidity [29,30].

In our study, mowing was defined as cutting the vegetation and immediately removing the biomass, which is different to real-life agricultural practices. Farmers typically leave the biomass remains on the surface for days, while it is dried, turned, windrowed, and finally baled and removed. Leaving the cut hay temporarily out might prevent desiccation, thus reducing the negative effects of mowing. However, in nature conservation treatments focused on small areas, the immediate removal of the cut biomass is standard practice. Therefore, our study rather represents current nature conservation, than agricultural practices.

Careful selection of management conditions can significantly reduce their negative impact on flora and fauna. These include staggering treatments throughout the year or leaving parts of the habitat unmanaged [36,41]. The negative effects of treatments could also be prevented or reduced by tailoring them to the life cycle of invertebrates [22,23]. It may also be beneficial not to manage an area every year, but – beyond a certain reduction in intensity – the advancing natural succession can threat the survival of the whole habitat [9,34].

Our experimental study had promising results, but we are still far from having sufficient knowledge of the effects of grassland treatments. Our research was limited to one season, two sites and a comparison of three treatment methods. Short-term results are important because they can guide future treatments. On the other hand, to achieve sustainability, longer-term studies are needed and it would be useful to expand our study in multiple ways. First, by including additional litter or soil-dwelling taxa, in order to get a more complete idea of the effects of each treatment on entire ecosystems. Second, by comparing both mowing and mulching as well as their common combinations, since in practice, management methods are often applied together, resulting in potentially different effects than just one treatment alone. Third, including different spatiotemporal patterns and intensities of treatments would allow us to even better understand the effects of real-world agricultural practices.

To summarise, significantly more research is needed to properly understand the effects of grassland management on flora and fauna. This would allow us to optimise real-world agricultural practices to each habitat, ensuring the most suitable management practices are used and thus best preserving biodiversity.

## Conclusions

Through a controlled grassland management experiment, we examined the effects of mowing and mulching on the snail communities of wet meadows. Our results indicate that mulching is a suitable way for short-term interventions of grassland areas without the negative effects of mowing on snail communities. However, while data on the effects of different management methods after one year are useful, there is still a lack of long-term monitoring of grassland management, particularly regarding mulching on wet meadows. Initiating, supporting and maintaining more long-term research projects

is crucial. In addition to economic considerations, ecological aspects of meadow management must also be taken into account to ensure sustainability for the future.

## Supporting information

**S1 Table. Total number of living individuals in different treatment plots and periods in the two study area.**
(DOCX)

**S2 Table. Pairwise comparison of treatments before the treatments.** Posthoc tests from GLMMs separately fitted to each response variable (see the main text for details).
(DOCX)

**S3 Table. Changes occurring in different treatment plots due to treatment.** Posthoc tests from GLMMs separately fitted to each response variable (see the main text for details). Legend: Bold font indicates significant differences.
(DOCX)

## Acknowledgments

We are grateful to Tamás Székely, Jr. for his comments on the manuscript.

## Author contributions

**Conceptualization:** Roland Farkas, Miklós Bán, György Dudás, Zoltán Barta.

**Data curation:** Zoltán Barta.

**Formal analysis:** Zoltán Barta.

**Funding acquisition:** Roland Farkas, György Dudás, Zoltán Barta.

**Investigation:** Roland Farkas, Gergő Oláh.

**Methodology:** Roland Farkas, Miklós Bán, Zoltán Barta.

**Project administration:** Roland Farkas, Zoltán Barta.

**Supervision:** Zoltán Barta.

**Validation:** Zoltán Barta.

**Visualization:** Roland Farkas, Miklós Bán, Gergő Oláh, György Dudás, Zoltán Barta.

**Writing – original draft:** Roland Farkas.

**Writing – review & editing:** Roland Farkas, Miklós Bán, Gergő Oláh, György Dudás, Zoltán Barta.

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
