## [Decision Letter · Decision Letter 0]

PONE-D-24-52289Experimental comparison of mowing and mulching on snail communities in wet meadowsPLOS ONE

Dear Dr. Farkas,

Thank you for submitting your manuscript to PLOS ONE. After careful consideration, we feel that it has merit but does not fully meet PLOS ONE’s publication criteria as it currently stands. The study addresses an important and underexplored topic but requires substantial improvements in its methodology, data analysis, and presentation. Therefore, we invite you to submit a revised version of the manuscript that addresses the points raised during the review process.

We look forward to receiving your revised manuscript.

Kind regards,

Edvard Mizsei

Academic Editor

PLOS ONE

Journal Requirements: When submitting your revision, we need you to address these additional requirements. 1. Please ensure that your manuscript meets PLOS ONE's style requirements, including those for file naming. The PLOS ONE style templates can be found at https://journals.plos.org/plosone/s/file?id=wjVg/PLOSOne_formatting_sample_main_body.pdf and https://journals.plos.org/plosone/s/file?id=ba62/PLOSOne_formatting_sample_title_authors_affiliations.pdf 2. In your Methods section, please provide additional information regarding the permits you obtained for the work. Please ensure you have included the full name of the authority that approved the field site access and, if no permits were required, a brief statement explaining why.

**Additional Editor Comments:**

The study’s limited scale (two sites, three treatments) necessitates a careful acknowledgement of its constraints.

Include limitations of the study and suggestions for future research, particularly regarding long-term effects and additional taxa.

Provide precise details about the mowing and mulching machinery used (e.g., bar mower, disc mower, or flail mulcher).

Address awkward phrasing and inconsistent terminology (e.g., replace "parcel" with "plot," and "vegetable origin" with "organic material").

Differentiate between immediate and longer-term effects of management practices and contextualize results within this framework.

Reviewers' comments:

Reviewer's Responses to Questions

**Comments to the Author**

1. Is the manuscript technically sound, and do the data support the conclusions?

Reviewer #1: Yes

Reviewer #2: Yes

2. Has the statistical analysis been performed appropriately and rigorously? 

Reviewer #1: Yes

Reviewer #2: Yes

3. Have the authors made all data underlying the findings in their manuscript fully available?

Reviewer #1: Yes

Reviewer #2: No

4. Is the manuscript presented in an intelligible fashion and written in standard English?

Reviewer #1: No

Reviewer #2: Yes

5. Review Comments to the Author

Reviewer #1: This study investigates the effects of two different grassland management techniques—mowing and mulching—on mollusk communities in two wet meadow sites. The aim was to determine which management practice would have the least detrimental impact on these communities. While not explicitly stated, the authors anticipated that the management practices would have varying effects on the number of live mollusk specimens, species richness, and Shannon diversity. The authors found that mowing have significant effect on the mollusk communities: mowing reduced the number of living individuals and number of species, but increased Shannon diversity. Mulching had no significant effect. The study concludes that grassland management practices can alter mollusk communities in wet meadows, highlighting the importance of considering these impacts when choosing management strategies.

The authors' experimental design for this field study is commendable. Despite the small scale, the research addresses a relevant question regarding conservation management, particularly given the scarcity of literature on the effects of mulching on grassland fauna. A key weakness of the study design is the inconsistency in sampling times between years. While pre-treatment data were collected in June, post-treatment data were collected in August of the following year. The study's short-term before-after design, without subsequent monitoring in the years following treatment application, restricts its ability to assess long-term community responses and potential delayed effects of mowing and mulching. While the manuscript generally meets the formal requirements of a scientific paper, the language requires substantial improvement. Numerous typos, awkward phrasing, and occasional illogical sentence structures suggest potential translation issues.

Abstract:

line19: ...maintained through appropriate management method …

line21: shredding, instead of fragmentation

line22: ...is becoming more widely practiced. Or something similar

line22-23: Informal. It could be replaced with more precise academic language.

line30-31: ...while also preserving their biodiversity.

Introduction:

line42: use in-text reference consistently. Suggestion: Squires et al. 2018

line40-41: The phrase "necessary for, first the nomadic lifestyle of livestock grazing" is awkwardly phrased and could be restructured for smoother readability.

line43-44: suggest: since the agricultural revolution, primarily due to human activities

line62: utilizing it, rather than farming

line64-65: Rephrase this sentence, expressions and grammar!

Line73-74: Consider breaking it into two sentences for clarity.

Line76-78: I do not think such explanation is needed for the practices.

Line79-81: Suggestion: The encroachment of shrubs and woody plants threatens the persistence of grasslands and can also negatively impact farming activities.

Line81: A striking example of these changes can be seen in Central and Eastern Europe…

Line85: use ; in the parentheses between 2 references

Line86: consistency in in-text references still needed

Line88: definitions, rather than meanings

Line96: lawn is a wrong translation of grassland/meadow

Line100: rather an area within a few years

Line109: “Less favourable temperatures" could be more specific, mentioning higher temperatures

Line114: Organic material/Biomass

Line132: ...has been less extensively studied.

Line124: do not use the word “farmed” in this context

Line151-152: The phrase "At most, conclusions can be drawn from experience with one kind of mulching" sounds awkward and informal.

Line155: populations can’t be indicator species

Line156: "to say after the first year" could be more formal

Line158-159: Suggest: We selected undisturbed wet meadows that had remained untouched for decades and characterized their snail communities. We then mowed and mulched portions of these areas using standard agricultural machinery, following a stratified experimental design.

In the aims section, you could be more precise, mention at least hte factors, which you use to assess how “harmful” these managements are. Hypotheses could be clarified as well.

Materials and methods:

Line175: I would avoid the phrasing “without any obvious microhabitat structure”, as microhabitat structure depends of the definition. Be precise.

Line177: “for the last 20-30 years” 10 years matters a lot!

Line180: Figure 1.: In my opinion, the two smaller maps below are not very informative. Consider replacing them with a different figure. It's important to show the location within Hungary, but additionally, it might be useful to show Hungary's location within Europe, rather than using two other small-scale maps which shows nothing. Including satellite images to illustrate vegetation coverage would be helpful, and adding some toponyms could assist those who want to identify the study sites. Alternatively, you could completely reorganize the figure. Besides the map of Hungary, photographs of the meadows might be more useful for better understanding.

Line196: It would be more obvious to show 45 sampling points on Figure 2, rather than a grid.

Line204: don’t need “once”

Line207-208: I think the aouthors did not mimic the traditional mowing of hay this way. The cut plant material remaines on the are for some days for drying. This is a totally different effect on snails, then removing the plant material immediately.

Line222: “vegetable origin”:)

Line241: don’t need “interactive”

Line253: don’t need “however”

Results:

Line276: scientific names should be italic in Table 1

Line292: authors may use Shannon diversity consitently

Line305-306: indexing χ21 χ22 χ22

On Figure 3c, relocate y axis title

Discussion:

Line328: check this sentence

Line 389: there is in additional comma in the reference parentheses

I would like to read some limitations, possible future prospects in the discussion. This part was much better formulated than the other parts of the article.

Conclusion:

Line403: avoid such strong emotional phrases like “clearly evidendt”

Line405-406: Be cautious with the word “ptactical”. Highlight the exact result like:...without the negative effects of mowing on snail communities

Line407: “to maintain” and “to preserve” would fit better

Line407-408: Link it more directly to your findings. Like: “Given the observed negative effects of mowing…”

Line413-415: This is too general. Connect it to the specific ecological aspects revealed by the study.

Literature:

Again, be consistent. Corrected author initials spacing (e.g., "KJ." → "K. J."), it is used multiple ways.

Appendix 2: Format the first column, odd spacing.

Reviewer #2: The manuscript “ Experimental comparison of mowing and mulching on snail communities in wet meadows“ by Farkas et al. studied the effect of two common grassland management methods on the number of individuals, the species numbers and the diversity (Shannon-Index) of snails in wet meadows in Hungary one year after the treatment.

While mulching, the shredding of vegetation and leaving it on the ground, had no effect, mowing reduced the number of individuals and species, but increased the Shannon-diversity. From the most abundant species, only one was significantly affected by the treatment, with an increase in the number of individuals after mulching.

Technically, the study seems to be sound and the results seem to be well supported by the data. Samples were collected with appropriate methods and the data were analysed using up-to-date statistical methods. Therefore, considering the lack of data on the effect of intensive agriculture on snails in the face of the current biodiversity crisis the data certainly deserve publication.

My main concern is that the analysis of data is a bit meager and that the study is relatively small and consisted of one experiment only with three treatments (control, mowing, mulching) on two sites. While the first aspect can be addressed by some more statistical analyses (see remarks below), the second cannot be changed. Therefore, I would leave it to the editor to decide if this is sufficient to accept the manuscript for publication. Apart from this, I have several aspects that should be addressed to make the manuscript suitable for publication.

Detailed comments

1. While it is interesting that only mowing affected individual numbers, species numbers and Shannon-diversity, it would be interesting to know the underlying reasons. This is the case for the demonstrated effects of mowing, but also the absence of any effect of mulching.

For instance, the unexpected rise in the Shannon-diversity by mowing is explained by the authors by a potential decline of individual numbers of abundant species and the associated increase of evenness. This is an interesting hypothesis which could (and should) be tested by the authors with their data. First, authors should calculate the evenness for their samples. In addition, a closer look at the data should reveal whether there are in fact abundant species which have decreased in individual numbers. Likewise, it would be interesting to know, whether mulching really had no impact on the snail community (as suggested by the numbers of individuals and species), or whether there might have been a species turn-over between the plots before and after the treatment. This could be analysed using beta-diversity and a similarity analysis. The same analysis should be done with the data from the mown plots. In addition, authors should also look at the species lists from the different treatments and the ecology of species that have changed in abundance. In conclusion, a more detailed analysis of the data could over insights into the mechanisms which are responsible for the observed effects.

2. While it is correct that the authors have included “site” as random variable in their model, it would be also interesting to learn whether the observed effects occur equally on both sites. Therefore, I suggest an analysis in which site is used as fixed factor.

3. Generally, the manuscript is well readable but not very well written and, in some parts, not very stringent and wordy. This is especially true for the introduction and the discussion. Also, the English sometimes seems wrong to me. For instance, the term “parcel” is used for the German term “Parzelle”. I think the correct translation should be “plots”. I have made a number of suggestions in the attached pdf-File, but I am not a native speaker. Therefore, I strongly recommend proof reading of the manuscript by a native speaker experienced in scientific writing before submission of the revised version.

4. Authors should not refer to snails as indicator species in the context of the study. As mentioned by the authors themselves, different taxa react very differently to agricultural management measures. Snaily might be indicators for themselves, but not for other groups!

5. Authors should be more specific with respect to the effect of mowing or mulching on the target organisms. This refers to line 133 of the introduction, but also to the rest of the manuscript. In the available literature, the effect of mowings has been studied immediately after the treatment, sometime after the treatment (weeks or years), or during longer periods of time. While effects found immediately after the treatment have to (and can) be attributed directly to the mowing event itself, i.e. the mortality caused by the mowing machine, effects after weeks or years can be increasingly indirect, e.g. increased mortality due to unfavourable environmental conditions or increased predation of exposed organisms. Authors should explicitly point out that their study refers to the second effects

6. On how many replicates per treatement are the data based. Is it n=15 per site, i.e. n=30 per treatment for the whole study?

7. Throughout the ms: Tables description should be on top of the table.

Line 93ff: What is the difference between these two variants of mulching. To me, it sounds identical. Grass is mown and left on the spot. Or is the fact that grass is shredded in variant 3, but not in variant 2?

Line 207: Please include the type of mowing machine used? Bar mower, Disc mover

Flail mulcher? Conditioner?

Table 2: Use italics for species and genus names

6. PLOS authors have the option to publish the peer review history of their article (what does this mean? ). If published, this will include your full peer review and any attached files.

**Do you want your identity to be public for this peer review?** For information about this choice, including consent withdrawal, please see our Privacy Policy .

Reviewer #1: No

Reviewer #2: **Yes: ** Johannes Steidle

---

## [Author Response · Author response to Decision Letter 1]

20 May 2025

Dear Reviewers,

I would like to thank you for your dedicated work, which contributed to the development of our manuscript. We found all your questions and suggestions valuable. We have done our best to revise the manuscript with your help in accordance with the professional needs of the scientific field and the journal. We especially thank you for your patience and understanding regarding the linguistic shortcomings, upon your suggestion we had it checked by a native speaker. We have responded to all your questions and suggestions, which you can find in the uploaded rebuttal letter. We thank you once again for your work and wish you further success in your professional career.

Additional Editor Comments:

The study’s limited scale (two sites, three treatments) necessitates a careful acknowledgement of its constraints.

Thank you for your comment, we have included your suggestion in the Discussion chapter.

Include limitations of the study and suggestions for future research, particularly regarding long-term effects and additional taxa.

We have made the requested additions int he Discussion chapter..

Provide precise details about the mowing and mulching machinery used (e.g., bar mower, disc mower, or flail mulcher).

The parameters were defined in the Method section upon your request.

Address awkward phrasing and inconsistent terminology (e.g., replace "parcel" with "plot," and "vegetable origin" with "organic material").

Based on your and the reviewers' helpful and detailed suggestions, we have made several modifications to the wording of our manuscript. We are confident that the readability of the manuscript has improved.

Differentiate between immediate and longer-term effects of management practices and contextualize results within this framework.

We have increased the elaboration of the question in both the Introduction and the Discussion chapters.

Reviewers' comments:

Reviewer's Responses to Questions

Comments to the Author

1. Is the manuscript technically sound, and do the data support the conclusions?

Reviewer #1: Yes

Reviewer #2: Yes

2. Has the statistical analysis been performed appropriately and rigorously?

Reviewer #1: Yes

Reviewer #2: Yes

3. Have the authors made all data underlying the findings in their manuscript fully available?

Reviewer #1: Yes

Reviewer #2: No

4. Is the manuscript presented in an intelligible fashion and written in standard English?

Reviewer #1: No

Reviewer #2: Yes

5. Review Comments to the Author

Reviewer #1: This study investigates the effects of two different grassland management techniques—mowing and mulching—on mollusk communities in two wet meadow sites. The aim was to determine which management practice would have the least detrimental impact on these communities. While not explicitly stated, the authors anticipated that the management practices would have varying effects on the number of live mollusk specimens, species richness, and Shannon diversity. The authors found that mowing have significant effect on the mollusk communities: mowing reduced the number of living individuals and number of species, but increased Shannon diversity. Mulching had no significant effect. The study concludes that grassland management practices can alter mollusk communities in wet meadows, highlighting the importance of considering these impacts when choosing management strategies.

The authors' experimental design for this field study is commendable. Despite the small scale, the research addresses a relevant question regarding conservation management, particularly given the scarcity of literature on the effects of mulching on grassland fauna. A key weakness of the study design is the inconsistency in sampling times between years. While pre-treatment data were collected in June, post-treatment data were collected in August of the following year.

Thank you for your comment. The pre-management data collection, mechanical management and post-treatment data collection were performed simultaneously in the control, mowed and mulched plots. Consequently, the same amount of time elapsed between each action in all three treatment plots, therefore we believe that the results of our tests (a significant interaction between treatment and sampling period) were not affected by the fact that it covered a period of 14 months rather than a calendar year. We agree that the terms "short term" or "one year" used for the duration of our work are not accurate, so we have corrected this in the manuscript.

The study's short-term before-after design, without subsequent monitoring in the years following treatment application, restricts its ability to assess long-term community responses and potential delayed effects of mowing and mulching.

Our design was specifically to examine the short-term effects of treatments. We tried to avoid to draw long-term conclusions based on our results. Changes occurring in the short term may help to identify adverse processes early, which allows for the transition to other, more appropriate methods in sensitive areas. We also described this idea in the manuscript.

While the manuscript generally meets the formal requirements of a scientific paper, the language requires substantial improvement. Numerous typos, awkward phrasing, and occasional illogical sentence structures suggest potential translation issues.

We are very grateful that you devoted so much energy to improving the language of our manuscript! Your suggestions meant a lot to us, so we used them in almost every case to improve the manuscript. We also asked a fluent English speaker to improve the language of our MS.

Abstract:

line19: ...maintained through appropriate management method …

line21: shredding, instead of fragmentation

line22: ...is becoming more widely practiced. Or something similar

line22-23: Informal. It could be replaced with more precise academic language.

line30-31: ...while also preserving their biodiversity.

Introduction:

line42: use in-text reference consistently. Suggestion: Squires et al. 2018

line40-41: The phrase "necessary for, first the nomadic lifestyle of livestock grazing" is awkwardly phrased and could be restructured for smoother readability.

line43-44: suggest: since the agricultural revolution, primarily due to human activities

line62: utilizing it, rather than farming

line64-65: Rephrase this sentence, expressions and grammar!

Line73-74: Consider breaking it into two sentences for clarity.

Line76-78: I do not think such explanation is needed for the practices.

Line79-81: Suggestion: The encroachment of shrubs and woody plants threatens the persistence of grasslands and can also negatively impact farming activities.

Line81: A striking example of these changes can be seen in Central and Eastern Europe…

Line85: use ; in the parentheses between 2 references

Line86: consistency in in-text references still needed

Line88: definitions, rather than meanings

Line96: lawn is a wrong translation of grassland/meadow

Line100: rather an area within a few years

Line109: „Less favourable temperatures" could be more specific, mentioning higher temperatures

Line114: Organic material/Biomass

Line132: ...has been less extensively studied.

Line124: do not use the word „farmed” in this context

Line151-152: The phrase "At most, conclusions can be drawn from experience with one kind of mulching" sounds awkward and informal.

Line155: populations can’t be indicator species

Line156: "to say after the first year" could be more formal

Line158-159: Suggest: We selected undisturbed wet meadows that had remained untouched for decades and characterized their snail communities. We then mowed and mulched portions of these areas using standard agricultural machinery, following a stratified experimental design.

In the aims section, you could be more precise, mention at least hte factors, which you use to assess how „harmful” these managements are. Hypotheses could be clarified as well.

Thank you for your comment, we have improved the sections indicated above according to your request.

Materials and methods:

Line175: I would avoid the phrasing „without any obvious microhabitat structure”, as microhabitat structure depends of the definition. Be precise.

Line177: „for the last 20-30 years” 10 years matters a lot!

Line180: Figure 1.: In my opinion, the two smaller maps below are not very informative. Consider replacing them with a different figure. It's important to show the location within Hungary, but additionally, it might be useful to show Hungary's location within Europe, rather than using two other small-scale maps which shows nothing. Including satellite images to illustrate vegetation coverage would be helpful, and adding some toponyms could assist those who want to identify the study sites. Alternatively, you could completely reorganize the figure. Besides the map of Hungary, photographs of the meadows might be more useful for better understanding.

We have transformed the figure. We considered the photographic solution to be appropriate, which was indeed more informative than the original. We solved the identification of the locations using coordinates given in the Method section.

Line196: It would be more obvious to show 45 sampling points on Figure 2, rather than a grid.

The figure has been modified and incorporated into Figure 1.

Line204: don’t need „once”

Line207-208: I think the aouthors did not mimic the traditional mowing of hay this way. The cut plant material remaines on the are for some days for drying. This is a totally different effect on snails, then removing the plant material immediately.

Thank you for your comment. To clarify the issue, we have explained the difference between the treatment methods we use and the traditional ones in the Discussion section. It is interesting that the articles based on mowing experiments only mention the fact of "mowing". After cutting the vegetation, they do not detail how many sub-steps were performed (drying, turning, windrowing, baling, etc.) until the cut biomass was removed. Perhaps more emphasis should be placed on detailing these in future studies. In our modification we also highlighted that management practice of nature conservation areas by mowing resembles more to our treatment than traditional hay producing mowing.

Line222: „vegetable origin”:)

Line241: don’t need „interactive”

Line253: don’t need „however”

Results:

Line276: scientific names should be italic in Table 1

Line292: authors may use Shannon diversity consitently

Line305-306: indexing χ21 χ22 χ22

On Figure 3c, relocate y axis title

Discussion:

Line328: check this sentence

Line 389: there is in additional comma in the reference parentheses

I would like to read some limitations, possible future prospects in the discussion. This part was much better formulated than the other parts of the article.

The indicated parts were corrected in the manuscript at your request.

Conclusion:

Line403: avoid such strong emotional phrases like „clearly evidendt”

Line405-406: Be cautious with the word „ptactical”. Highlight the exact result like:...without the negative effects of mowing on snail communities

Line407: „to maintain” and „to preserve” would fit better

Line407-408: Link it more directly to your findings. Like: „Given the observed negative effects of mowing…”

Line413-415: This is too general. Connect it to the specific ecological aspects revealed by the study.

We modified this part at your request.

Literature:

Again, be consistent. Corrected author initials spacing (e.g., "KJ." → "K. J."), it is used multiple ways.

Thank you for your comment. We checked and corrected spacings throught the whole manuscript.

Appendix 2: Format the first column, odd spacing.

We modified this part at your request.

Reviewer #2: The manuscript „ Experimental comparison of mowing and mulching on snail communities in wet meadows“ by Farkas et al. studied the effect of two common grassland management methods on the number of individuals, the species numbers and the diversity (Shannon-Index) of snails in wet meadows in Hungary one year after the treatment.

While mulching, the shredding of vegetation and leaving it on the ground, had no effect, mowing reduced the number of individuals and species, but increased the Shannon-diversity. From the most abundant species, only one was significantly affected by the treatment, with an increase in the number of individuals after mulching.

Technically, the study seems to be sound and the results seem to be well supported by the data. Samples were collected with appropriate methods and the data were analysed using up-to-date statistical methods. Therefore, considering the lack of data on the effect of intensive agriculture on snails in the face of the current biodiversity crisis the data certainly deserve publication.

My main concern is that the analysis of data is a bit meager and that the study is relatively small and consisted of one experiment only with three treatments (control, mowing, mulching) on two sites. While the first aspect can be addressed by some more statistical analyses (see remarks below), the second cannot be changed. Therefore, I would leave it to the editor to decide if this is sufficient to accept the manuscript for publication. Apart from this, I have several aspects that should be addressed to make the manuscript suitable for publication.

We thanks the reviewer effort on our MS and reply her/his detailed comments below.

Detailed comments

1. While it is interesting that only mowing affected individual numbers, species numbers and Shannon-diversity, it would be interesting to know the underlying reasons. This is the case for the demonstrated effects of mowing, but also the absence of any effect of mulching.

We now highlight some reasons why mowing had and mulching had not have an effect on snail communities. These are basically connected to the fact that after mowing all cut material was removed from the area, while after mulching the material remained which prevented the immediate desiccation of the area.

For instance, the unexpected rise in the Shannon-diversity by mowing is explained by the authors by a potential decline of individual numbers of abundant species and the associated increase of evenness.

Here we have to point out a mistake we made in the previous version of the MS: We actually calculated Pileou’s evenness index for our samples which we incorrectly called the Shannon diversity. We now corrected this naming error throughout the MS.

This is an interesting hypothesis which could (and should) be tested by the authors with their data. First, authors should calculate the evenness for their samples. In addition, a closer look at the data should reveal whether

---

## [Decision Letter · Decision Letter 1]

PONE-D-24-52289R1The effect of mowing and mulching on snail communities: an experiment in wet meadowsPLOS ONE

Dear Dr. Farkas,

Thank you for submitting your manuscript to PLOS ONE. After careful consideration, we feel that it has merit but does not fully meet PLOS ONE’s publication criteria as it currently stands. Therefore, we invite you to submit a revised version of the manuscript that addresses the points raised during the review process.

Although you thoroughly revised your manuscript, both reviewers found further room for improvement.

We look forward to receiving your revised manuscript.

Kind regards,

Edvard Mizsei

Academic Editor

PLOS ONE

Journal Requirements:

Reviewers' comments:

Reviewer's Responses to Questions

**Comments to the Author**

1. If the authors have adequately addressed your comments raised in a previous round of review and you feel that this manuscript is now acceptable for publication, you may indicate that here to bypass the “Comments to the Author” section, enter your conflict of interest statement in the “Confidential to Editor” section, and submit your "Accept" recommendation.

Reviewer #1: All comments have been addressed

Reviewer #2: All comments have been addressed

2. Is the manuscript technically sound, and do the data support the conclusions?

Reviewer #1: Yes

Reviewer #2: Yes

3. Has the statistical analysis been performed appropriately and rigorously? 

Reviewer #1: Yes

Reviewer #2: Yes

4. Have the authors made all data underlying the findings in their manuscript fully available?

Reviewer #1: Yes

Reviewer #2: No

5. Is the manuscript presented in an intelligible fashion and written in standard English?

Reviewer #1: Yes

Reviewer #2: Yes

6. Review Comments to the Author

Reviewer #1: The revised manuscript shows clear improvement, and it is evident that the authors have carefully addressed the reviewers’ earlier comments. The text reads more smoothly, the structure and clarity have been improved, and the scientific content is now better communicated. I appreciate the thoughtful revisions and the effort made to respond comprehensively to the feedback. Below are some further suggestions for fine-tuning the manuscript.

27-28: I would highlight the relevance in conservation, e.g.: Therefore, mulching may be a promising candidate for conservation management in wet meadow habitats.

51-52: The end of this sentence is not clear. Should use something to clarify, like ....making this a critical conservation concern.

65: modern?

73: I would insert "management" before strategy.

170: In this case, you should write out the English name of the species in my opinion, or just the scientific name, without parentheses.

172: I understand, what would you like to say with this sentence, but it is formulation isn't clear. Something like "Both sites were flat and appeared to have a homogeneous vegetation structure." would be clear.

194: You don't need the "." after years.

209: The "X" is capitalised in 25x25, change it please.

314-315: Clunky sentence sructure. I would rather choose this: Only for Vertigo angustior did the treatments have significantly different effects on abundance (interaction: ?22 = 7.90, p = 0.019);

326: ...but not in mulched ones,

400: no need for ","

On Figure 2, the species names should be written with italic in my opinion.

Reviewer #2: Generally, the manuscript was improved a lot although it is still very wordy and repetitious. I have made some suggestions in the attached pdf and I think that including these corrections, it is suitable for publication.

7. PLOS authors have the option to publish the peer review history of their article (what does this mean? ). If published, this will include your full peer review and any attached files.

**Do you want your identity to be public for this peer review?** For information about this choice, including consent withdrawal, please see our Privacy Policy .

Reviewer #1: No

Reviewer #2: No

---

## [Author Response · Author response to Decision Letter 2]

22 Jun 2025

Dear Reviewers,

Thank you for your valuable comments that contributed to the development of our manuscript. In the second round of review, both of you made further suggestions for improving the language and style of the manuscript. In each case, your suggestions were justified and important. We have accepted all of your comments and modified the manuscript. Thank you again for your contribution. The work of both of you has greatly helped us to formulate our results in the appropriate scientific form and style.

---

## [Editor Report · Decision Letter 2]

The effect of mowing and mulching on snail communities: an experiment in wet meadows

PONE-D-24-52289R2

Dear Dr. Farkas,

We’re pleased to inform you that your manuscript has been judged scientifically suitable for publication and will be formally accepted for publication once it meets all outstanding technical requirements.

Kind regards,

Edvard Mizsei

Academic Editor

PLOS ONE
---

## [Editor Report · Acceptance letter]

PONE-D-24-52289R2

PLOS ONE

Dear Dr. Farkas,

I'm pleased to inform you that your manuscript has been deemed suitable for publication in PLOS ONE. Congratulations! Your manuscript is now being handed over to our production team.

Kind regards,

on behalf of

Dr. Edvard Mizsei

Academic Editor

PLOS ONE